# Ginkgo biloba extract increases neurite outgrowth and activates the Akt/mTOR pathway

**Imane Lejri[1,2], Amandine Grimm[1,2], Anne Eckert[1,2]***

**1** University of Basel, Neurobiology Lab for Brain Aging and Mental Health, Transfaculty Research Platform Molecular and Cognitive Neuroscience, Basel, Switzerland, **2** Psychiatric University Clinics, University of Basel, Basel, Switzerland

\* anne.eckert@upk.ch

## Abstract

### Background

Standardized Ginkgo biloba extract (GBE) has demonstrated efficacy in the cognitive functional neuropsychiatric symptoms of patients with Alzheimer's disease (AD). With regard to its underlying molecular mode of action, first evidence was provided that GBE was able to modulate neuronal outgrowth *in vitro*, but the mechanisms underlying GBE effects on neuroplasticity remain unclear.

### Methodology/Principal findings

In this study, we investigated the effect of GBE on neurite outgrowth using SH-SY5Y neuroblastoma cells in a 2D and 3D surface culture. The effects of the GBE LI1370 on neuroplasticity and neurite outgrowth were compared to those of nerve growth factor (NGF, 50 ng/ml) which was used as a positive control. We evaluated several parameters of neurite outgrowth such as the neurite number, total neurite length and extend of branching. Our findings showed that GBE (10 and 100 µg/ml) significantly increased neurite outgrowth in the 2D as well as 3D culture model after 3 days of treatment with a comparable effect than that NGF. The use of the 3D cell culture allowed us to better reproduce the *in vivo* neuronal microenvironment for the evaluation the neurite formation after GBE treatment. In addition, we assessed the effects of GBE on the Akt/mTOR pathway, which is known to promote neuroplasticity induced by nerve growth factors. We showed that GBE treatment induced an increase of phosphorylated IGF1R (Tyr1135/Tyr1136), Akt (Ser473), TSC2 (Ser939), mTOR (Ser2448), PTEN (Ser380) and GSK3β (Ser9).

### Conclusion

Together, these findings indicate that GBE promotes neurite growth and activates the PI3K/Akt/mTOR pathway suggesting that this plant extract supports neuronal plasticity.

**Data Availability Statement:** All relevant data are within the paper and supporting information files.

**Funding:** This study was partly supported by a principal investigator (AE) - initiated research grant supported by Vifor Pharma Switzerland, with

regard to consumables and materials as well as GBE supply. Remaining funding was provided by grants from the Swiss National Science Foundation (SNF #31003A_149728, to AE) and Synapsis Foundation - Alzheimer Research Switzerland ARS to AG, and funding from the Transfaculty Research Platform, Molecular & Cognitive Neuroscience, University of Basel. The funders had no role in study design, data collection and analysis, decision to publish, or preparation of the manuscript.

**Competing interests:** AE received study grants from Vifor Pharma Switzerland. Moreover AE and AG received lecturer/consultant fees from Vifor Pharma Switzerland. This does not alter our adherence to PLOS ONE policies on sharing data and materials.

## Introduction

In accordance with the Pharmacopoea Europaea, the standardized Ginkgo biloba extract (GBE, LI1370) consists of 22.0–27.0% ginkgo flavone glycosides as well as 5.4–6.6% terpenoids. GBE has been shown to improve effectively mitochondrial defects through several modes of action such as antioxidant and the radical scavenging properties as well as amelioration of mitochondrial respiration and adenosine triphosphate (ATP) production [1–4]. The flavonoids (including isohammetin and kaempferol) seem to play an important role for free radical scavenging, whereas terpene lactones show substantial mitochondria-protecting properties [5]. Terpenes (including bilobalide and ginkgolides A,B,C) prevent membrane damage against free radicals and possess also several other neuroprotective properties. Flavonoids can act via neuronal receptors and modulate transcription factors, kinase signalling pathways and protein expression related to learning process and memory as well as cell proliferation [6]. Previously, we have investigated the protective effects of the extract LI 1370 on energy metabolism defects in human neuroblastoma cells (SH-SY5Y cells) [7]. GBE treatment (24hr, 100 μg/ml) was able to increase the coupling state of mitochondria leading to an amelioration of the efficiency of the mitochondrial electron transport chain (ETC). The increase of mitochondrial bioenergetics through the oxygen consumption and ATP production is due to the modulation of mitochondrial complex I, III and IV activities. Moreover, GBE treatment induced also an increase in the mitochondrial DNA (mtDNA) content. Thus, we clearly highlighted in our previous report the beneficial effect of GBE on mitochondrial function and energy metabolism [7]. Mitochondria are central regulators of fundamental processes in neuroplasticity, including neurite outgrowth [8]. Neurite outgrowth is a process where forming neurons produce new projections as they grow in response to guidance cues. Nerve growth factor (NGF), brain-derived neurotrophic factor (BDNF) or neurotrophins, are examples of such stimuli that regulate neurite growth. Müller and colleagues already demonstrated preliminary evidence that the standardized Gingko biloba extract EGb761® is able to increase the length of dendrites in a cell line derived from a pheochromocytoma of the rat adrenal medulla (PC12 cells) [9] but the underlying pathways of GBE effect on the neurite outgrowth are still unclear.

For this purpose, we first characterized the effect of GBE on neurite outgrowth in differentiated SH-SY5Y neuroblastoma cells using standard two-dimensional (2D) and three-dimensional (3D) cellular culture models. Then, we investigated the intracellular signal transduction pathways involved in promoting the neuroplasticity which is targeted by GBE. Overall, the data reported in the present study provide new insights into the molecular mechanisms of neurite extension induced by GBE, highlighting new potential therapeutic targets.

## Material and methods

### Chemicals and reagents

Dulbecco's-modified Eagle medium (DMEM), fetal calf serum (FCS), penicillin/streptomycin, neurobasal medium, and retionic acid were from Sigma-Aldrich (St. Louis, MO, USA). Gluta-max and B27 supplement were from Gibco Invitrogen (Waltham, MA, USA). NGF was from Lubio (Zürich, Switzerland). Standardized Ginkgo biloba extract (GBE) LI 1370 (composition: 22.0–27.0% flavone glycoside content and 5.4–6.6% terpene lactone content, DEV 35–67: 1, extractant: acetone 60% (V / V)) was produced and supplied by Vifor SA, Villars-sur-Glâne, Switzerland.

## Cell culture

Human SH-SY5Y neuroblastoma cells were grown at 37˚C in a humidified incubator chamber under an atmosphere of 7.5% $CO_2$ in DMEM supplemented with 10% volume/volume (v/v) heat-inactivated FCS, 2 mM Glutamax and 1% (v/v) penicillin/streptomycin. Cells were passaged 1–2 times per week, and plated for treatment when they reached 80–90% confluence [10]. This cell line is a "neuron-like" cellular model that is widely used in Neuroscience to study neuronal differentiation.

For 2D cell culture, cell plates were coated with collagen type I (Rat tail BD Bioscience) at 0.05 mg/ml.

For the 3D cell culture system, a BD PuraMatrix Peptide Hydrogel (BD Catalog #354250 packaged in one vial containing 1% solution (w/v) of purified synthetic peptide) was used. 5mg/ml of PuraMatrix (0.5%, 50 μl for a 96-well plate) were obtained by dilution in sterile $dH_2O$ and added to the surface of the well. Gelation was promoted by carefully and slowly adding medium to each well (100 μl for a 96-well plate). Plates were placed in an incubator for 60 minutes to complete the gelation of the PuraMatrix. The medium was changed twice over a period of 1 hour to equilibrate the growth environment to physiological pH. Cells were carefully added in each well at of concentration of 5000 cells/well.

## Treatment paradigm

SH-SY5Y neuroblastoma cells were plated in 96 wells plates (black with clear bottom) that were coated for the 2D or 3D cell culture. The next day, cell differentiation was induced by adding fresh neurobasal medium containing 1% fetal bovine serum and 10 μM retinoic acid (RA) for 3 days. Then, cells were treated either with 50 ng/ml of NGF (positive control) or with GBE at three different concentrations: 10, 100 and 500 ug/ml (concentrations selected on the basis of [7]). After 3 days of treatment, cells were fixed with 2% paraformaldehyde. All media were exchanged every 2 days to ensure the availability of growth factors and GBE components in the culture.

## Immunostaining

The protocol was used with 2D or 3D surface cultures of cells in plates. For 96-well black microplates with a clear bottom, it was possible to directly image the samples without transferring the gel to a glass slide. Immunolabeling of neurites was performed using an anti βIII-tubulin and Alexa Fluor 488-conjugated secondary antibody (fluorescence emission in the green wavelength).

## Microscopy and analysis (software)

Images were taken using an inverted confocal microscope (Leica Microsystems TCS SPE DMI4000) attached to an external light source for enhanced fluorescence imaging (Leica EL6000) with a 10x objective. Pinhole settings were chosen in such a way that axially all the cells were entirely present within the confocal volume. For the 2D culture method, one layer was taken. For the 3D culture method, z-stacks were generated (3–4 layers) to visualize the whole 3D network. The maximum intensity projection was then used for analysis that was performed using ImageJ (Neurophology plugin) software to evaluate parameters of neuroplasticity:

- Soma count

- Neurite count

- Total neurite length

- Difference (Δ, delta) of increase in neurite count under treatment

- Number of branching points = attachment point

- Number of contact points = endpoint

- Extent of neurite branching = ratio of attachment point number to ending point number

## Characterization of neuroplasticity pathways via targeted protein expression profiling

*Quantitative detection of each protein was performed using the Luminex technology*: The MILLIPLEX® MAP Akt/mTOR Phosphoprotein 11-plex Magnetic Bead Kit was used to detect changes in phosphorylated GSK3β (Ser9), Akt (Ser473), PTEN (Ser380), IGF1R (Tyr1135/Tyr1136), TSC2 (Ser939), and mTOR (Ser2448) in cell lysates using the Luminex® system. Of note, the total level of targeted proteins was also assessed after GBE treatment but no significant effect was detected (Akt/mTOR protein 11-plex Magnetic Bead Kit 96-well Plate). The detection assay is a rapid, convenient alternative to Western Blotting and immuno-precipitation procedures. Each kit has sufficient reagents for one 96-well plate assay. Lumi-nex® uses proprietary techniques to internally color-code microspheres with multiple fluorescent dyes. Through precise concentrations of these dyes, distinctly colored bead sets of 500 5.6 μm non-magnetic or 80 6.45 μm magnetic polystyrene microspheres can be created, each of which is coated with a specific capture antibody. After an analyte from a test sample is captured by the bead, a biotinylated detection antibody is introduced. The reaction mixture is then incubated with Streptavidin-PE conjugate, the reporter molecule, to complete the reaction on the surface of each microsphere. The microspheres are illuminated, and the internal dyes fluoresce, marking the microsphere set(s) used in a particular assay. A second illumination source excites PE, the fluorescent dye on the reporter molecule. Finally, high-speed digital-signal processors identify each individual microsphere and quantify the result of its bioassay based on fluorescent reporter signals.

*Immunoassay protocol for lysate samples*: 3 days after treatment with 100 μg/ml GBE, SH-SY5Y cell pellets were isolated and treated with lysis buffer (Millipore Corp., Billerica, USA) containing protease inhibitors. Cell debris was removed by centrifugation at 14 000 g for 15 min. Equal amounts of soluble protein lysates were analysed for the level of various phosphorylated proteins using MILLIPLEX MAP 11-plex Multi-Pathway Signalling Phosphoprotein kit on the Luminex 200 system (Millipore Corp.).

The multiplex assay was performed in a 96-well plate. The detailed procedure is as follows: the plate was wet with 50 μL assay buffer for 10 min and decanted. Then, the following solution was added: 25 μL beads, 25 μL assay buffer in blank wells, and 25 μL lysate at 0.8 mg/mL to sample or control wells. The plate was incubated O/N at 4°C. The beads were washed two times and 25 μL biotinylated detection Ab cocktail were added and the plate was incubated at RT for 1 hour. The beads were washed one time and 25 μL of streptavidin-phycoerythrin was added for a 15 min incubation at RT. 25 μL amplification buffer was added to the SAPE. SAPE/amplification buffer was removed and beads were suspended in 100 μL assay buffer. Plate reading was performed on Luminex instrumentation.

## Statistical analysis

Data are given as the mean ± SEM, normalized to the untreated control group (= 100%). Statistical analyses were performed using the Graph Pad Prism software. For statistical comparisons,

one-way ANOVA was used, followed by Dunnett's multiple comparison tests versus the control (CTRL, untreated neuroblastoma cells). For statistical comparisons of two groups, Student unpaired *t*-test was used. P values <0.05 were considered statistically significant.

# Results

## GBE improved the neurite outgrowth in 2D cell culture

To investigate the effect of GBE on neurite outgrowth, three different concentrations of GBE 10, 100 and 500 μg/ml were tested (based on our previous study [7]) on differentiated SH-SY5Y cells using the 2D cell culture method. Nerve Growth Factor (NGF, 50 pg/ml) which is known as promotor of cell growth and survival was used as positive control. The 2D pictures (one layer of cells) were taken with the confocal microscope and analysed with the ImageJ Neurophology software. After 3 days of treatment with GBE, the 100 μg/ml dose was the most efficient concentration to increase the neurite count to a similar extent than the NGF itself (about 77% increase when compared to the untreated cells (CTRL)) (Figs 1 and 2). In fact, after 3 days, the treatment with GBE at 10 μg/ml ameliorated the total neurite length (about 52% of increase), attachment point (up to 113% of increase) as well as the endpoint with 71% of improvements when compared to untreated SH-SY5Y cells (Fig 2). However, at the higher concentration of 100 μg/ml, GBE significantly ameliorated all the parameters with an increase

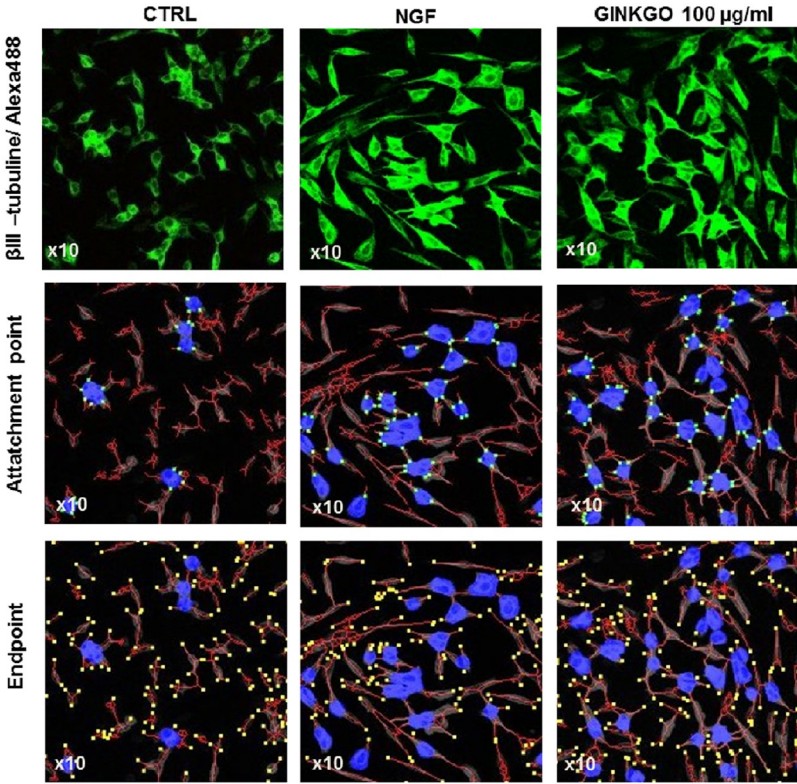

**Fig 1. GBE increased the neurite outgrowth of neuroblastoma cells after 3 days of treatment in a 2D cell culture.** Pictures were taken using a confocal microscope (x10). Immunostaining (IMS) with βIII- tubuline/Alexa488: pictures display neurite extension between the cells (upper panels; S1 Fig). Quantification of the neurite outgrowth parameters such as the attachment points (middle panels) and the endpoint numbers (lower panels), after NGF or GBE treatment are shown (Blue: soma, red: neurite, green point: attachment point, yellow point: endpoint).

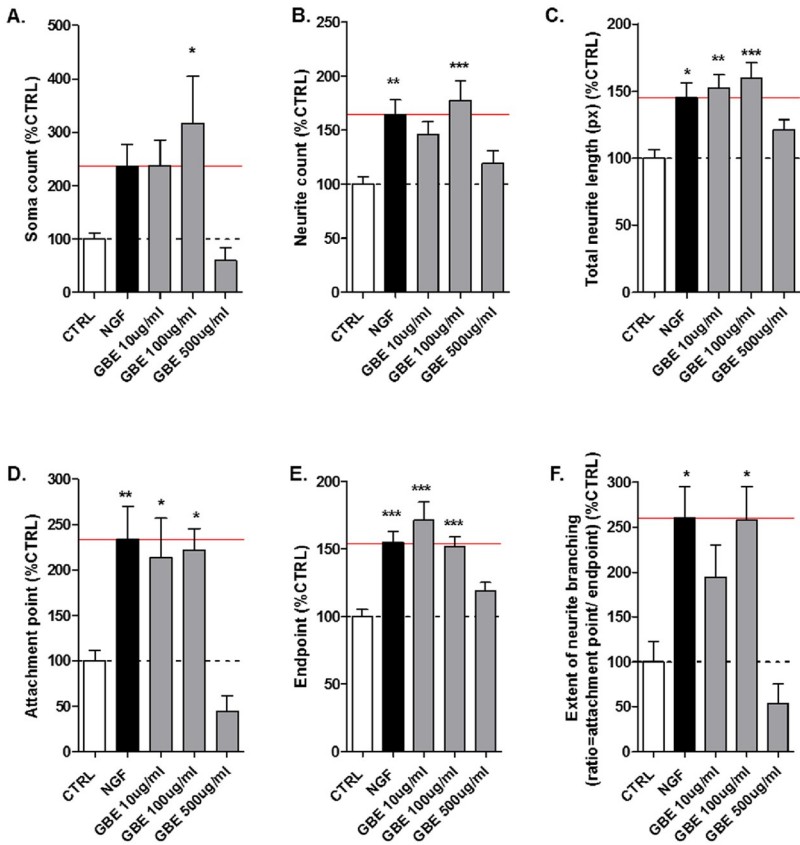

**Fig 2. GBE increased the neurite outgrowth of neuroblastoma cells after 3 days of treatment in a 2D cell culture.** Quantification of Fig 1 using NeurophologyJ (S1 Table). At 100 μg/ml GBE significantly increased: (A) cell number (soma count), (B) number of neurites (neurite count), (C) neurite length, (D) number of attachment points, (E) number of endpoints, and (F) extent of neurite branching. The effect of GBE was similar than the positive control NGF when compared to the untreated cells. The red line represents the effects of NGF-treated cells. Values represent the mean ±SEM of three independent experiments and were normalized to 100% of untreated CTRL cells. One way ANOVA and post hoc Dunnett's multiple comparisons versus untreated neuroblastoma cells (CTRL) *P<0.05,**P<0.01,***P<0.001.

going up to 216% for the neurite outgrowth with a similar effect than the positive control NGF (Figs 1 and 2) when compared to the untreated group. The higher concentration 500 μg/ml of GBE showed no significant effect on the neurite outgrowth in this model (Fig 2).

## GBE increased the neurite extension of cells in 3D-matrix

Based on the results obtained in the 2D cell culture system, we additionally evaluated the effect of the most effective concentration of GBE (100 μg/ml) using a 3D-Matrix system (Fig 3). In fact, we confirmed that GBE treatment (100 μg/ml) for 3 days ameliorated significantly neurite count (+79%), total neurite length (+75%), attachment (+138%) and endpoints (+64%) parameters and showed a positive trend to increase soma count and the extent of neurite branching compared to the untreated SH-SY5Y cells (Figs 3 and 4). Pictures obtained with the z-stack projection of 3–4 layers of cells on 3D-matrix show a 3D view of neurite outgrowth of the SH-SY5Y cells allowing the visualization of the formation of neurites and their projections between the cells (Fig 3).

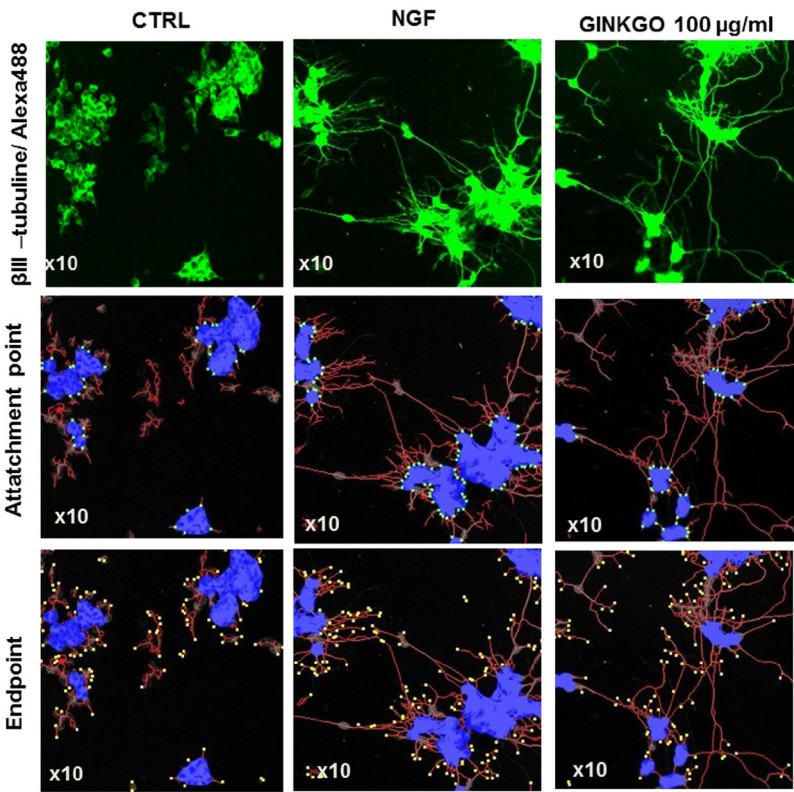

**Fig 3. GBE (100 μg/ml) induced neurite extension in a 3D-matrix by increasing the neurite outgrowth of neuroblastoma cells after 3 days of treatment.** Pictures were obtained by merging 3–4 layers of cells (z-stack projection) on 3D-matrix using a confocal microscope (x10). Immunostaining (IMS) with βIII- tubuline/Alexa488: pictures display neurite extension between the cells (upper panels, S2 Fig). Quantification of the neurite outgrowth parameters such as the attachment points (middle panels) and the endpoint numbers (lower panels), after NGF or GBE treatment are shown (Blue: soma, red: neurite, green point: attachment point, yellow point: endpoint).

## Targeted protein expression profiling induced by GBE treatment

Previous reports showed that GBE enhanced BDNF expression *in vitro* and *in vivo* [7, 11–13]. Because BDNF plays a role in neuroplasticity and neurite outgrowth, we decided to assess whether GBE was able to influence the BDNF signaling cascade by focusing on the Akt/mTOR pathway.

The Akt/mTOR signaling pathway is regulated by posttranslational modifications such as protein phosphorylation at multiple distinct sites, which are indicative of the activity state of the target protein. To characterize the effect of GBE (100 μg/ml, 3 days of treatment) on this pathway, quantitative detection of several proteins using Luminex technology kit was conducted (Fig 5). The MILLIPLEX® MAP Akt/mTOR phosphoprotein multiplex magnetic bead kit was used to detect changes in specific phosphorylated proteins in cell lysates. Compared to untreated cells, GBE (100 μg/ml) was able to increase significantly the level of the phosphoproteins Akt (Ser473), GSK3β (Ser9), IGF1R (Tyr1135/Tyr1136), mTOR (Ser2448), TSC2 (Ser939) and PTEN (Ser380) (Figs 5 and 6).

Thus, GBE seems to activate the Akt/mTOR pathway that supports cell survival and neurite outgrowth.

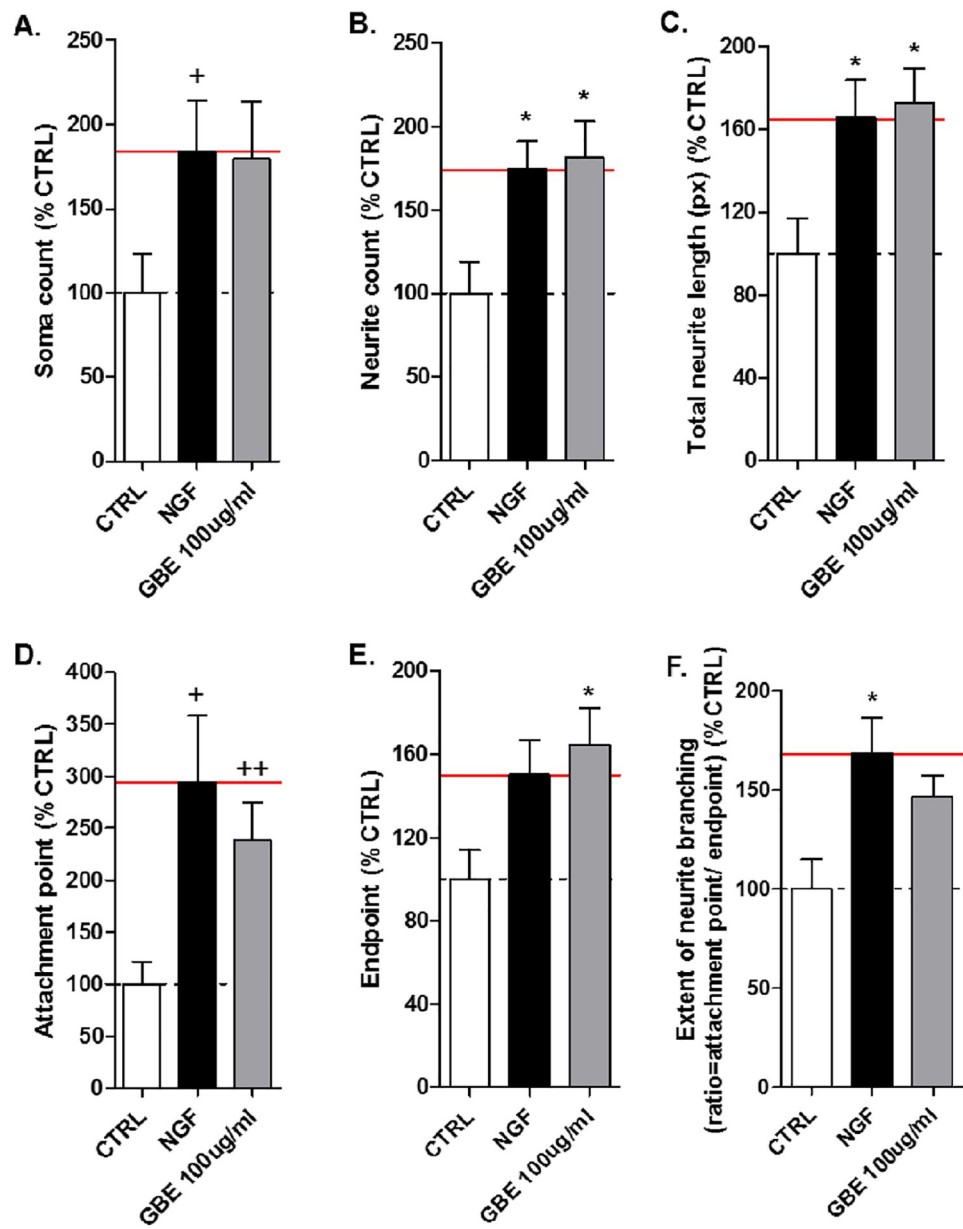

**Fig 4. GBE (100 μg/ml) increased the neurite outgrowth of neuroblastoma cells after 3 days of treatment using a 3D gel.** Quantification of Fig 3 using NeurophologyJ (S2 Table). GBE (100 μg/ml) ameliorated: (A) cell number (soma count), (B) number of neurites (neurite count), (C) neurite length, (D) number of attachment points, (E) number of endpoints, and (F) extent of neurite branching. The effect of GBE was similar than the positive control NGF when compared to the untreated cells. The red line represents the effects of NGF-treated cells. Values represent the mean ±SEM of three independent experiments and were normalized to 100% of untreated CTRL cells. One way ANOVA and post hoc Dunnett's multiple comparisons versus untreated neuroblastoma cells (CTRL) *$P<0.05$. Student t-test ⁺$P<0.05$, ⁺⁺$P<0.01$.

## Discussion

In this report, we demonstrated that the standardized GBE LI 1370 induced a significant increase of the neurite outgrowth in neuroblastoma cells after 3 days of incubation at 10 μg/ml and 100 μg/ml. These effects were comparable with the positive control NGF (50 ng/ml). In

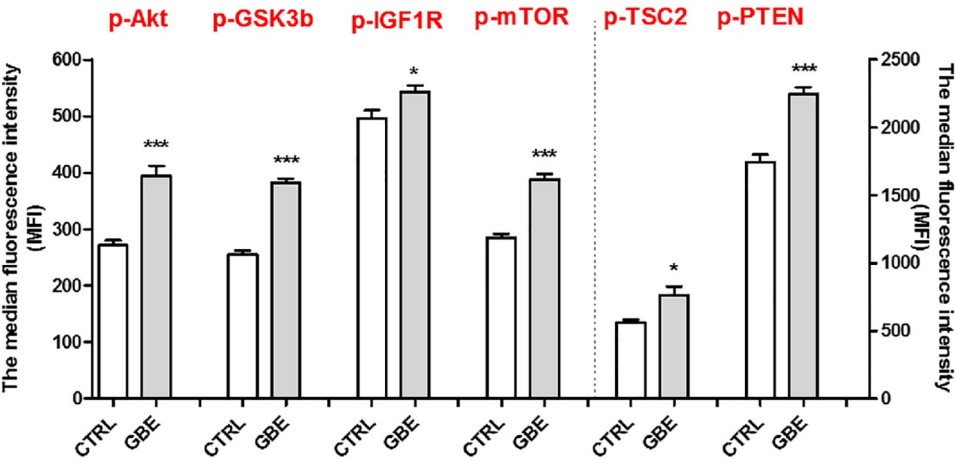

**Fig 5. Activation of the Akt/mTOR signaling pathway after GBE (100 µg/ml) treatment in neuroblastoma cells.**
Neuroblastoma cells (SH-SY5Y) stimulated with GBE (100 µg/ml) (3 days) were lysed in MILLIPLEX® MAP lysis buffer containing protease inhibitors. 20 µg total protein of each lysates diluted in MILLIPLEX® MAP assay buffer 2 were analyzed according the assay protocol (lysate incubation at 4˚C overnight). The Median Fluorescence Intensity (MFI) was measured measurements for the phosphoproteins Akt (Ser473), GSK3β (Ser9), IGF1R (Tyr1135/Tyr1136), mTOR (Ser2448), TSC2 (Ser939) and PTEN (Ser380) with the Luminex® system (S3 Table). The figure represents the average and standard deviation of three replicates. Student t-test *P<0.05, ***P<0.001.

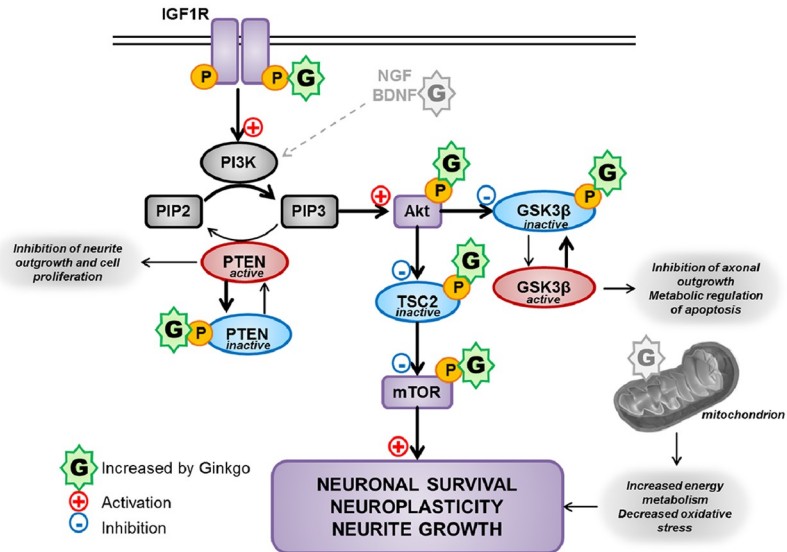

**Fig 6. Possible pathways involved in GBE-induced neuroplasticity pathway.** We propose that neuroprotective effects of GBE are mediated via multiple pathways. In the present study, we show that GBE increases IGF1R phosphorylation (Tyr1135/Tyr1136), which activates the Akt/mTOR pathway. Activation of Akt by phosphorylation (Ser473) inhibits TSC2 (phosphorylation at Ser939) which activates mTOR (phosphorylation at Ser2448) and induces neuronal survival, neurite outgrowth and neuroplasticity. In parallel, GBE inactivates the antagonizing pathways by increasing the phosphorylation of PTEN and GSK3β (at Ser380 and Ser9 respectively). Of note, the growth factors NGF and BDNF activate also the Akt/mTOR pathway, suggesting that GBE may also act via these growth factors to induce its neuroprotective effects. Besides, our previous study showed that GBE increases mitochondrial bioenergetics and decreases reactive oxygen species production (Rhein et al., PlosONE 2009), which may also play a role in neuroprotection and neurite outgrowth. Akt is also known as protein kinase B (PKB). IGF-1R, insulin-like growth factor-1; PI3K, phosphoinositide 3-kinase; PIP2 or PIP3, phosphatidylinositol triphosphate; PTEN, phosphatase and tensin homolog; GSK3, glycogen synthase kinase 3; TSC, tuberous sclerosis protein; mTOR, mammalian target of rapamycin.

line, another standardized GBE EGb761[®], enhanced neuritic outgrowth of PC12 cells at the same dose of 100μg/ml [9]. Among the three major constituents (ginkgolide, bilobalide, flavonoids), flavonoids seemed to be the most active group [9]. Moreover, Wang and Han showed that neurite outgrowth is also dramatically increased upon GBE treatment (50 μg/ml) in neural stem cells (NSCs) [14]. Besides, GBE treatment (50 μg/ml) significantly enhanced the formation and growth of neurosphere [14].

GBE-induced neurite outgrowth was compared in 3D-matrix of SH-SY5Y cultures using Puramatrix peptide hydrogel and in a standard 2D SH-SY5Y culture model using collagen I. In 3D culture, GBE (3 days, 100 μg/ml) showed even a better effect on neurite extension compared to the 2D SH-SY5Y culture.

NGF plays a crucial role in differentiation, survival and the stimulation of neurite outgrowth in neuronal cells [15]. The neurotrophic factor NGF has therefore been considered for treating the neurodegenerative diseases such as AD [16–19]. Neuroplasticity, the capacity of synapses to undergo structural adaptations in response to functional need or dysfunctions is deteriorated in aging and Alzheimer's disease [20]. Isorhamnetin, an aglycon of the flavonoids from *Ginkgo biloba* plant, was shown to be an inducer of neuroplasticity by increasing on the neurite length of the PC12 cells [9, 21]. Isorhamnetin was shown to induce a significant enhancement of neurotrophin (such as NGF)-mediated neurite outgrowth in PC12 cells [22]. Isorhamnetin effectively upregulated the production and release of NGF, BDNF and glial cell line-derived neurotrophic factor (GDNF) in primary rat astrocytes through estrogen signaling pathways [23]. Studies conducted using the standardized Gingko biloba extract (EGb761) showed that this extract enhances phosphorylation of cyclic-AMP Response Element Binding Protein (CREB) and increases the levels of phosphoCREB and BDNF, a member of the neurotrophin family of growth factors that plays a role in the proliferation, differentiation, and growth of neurons during development [12, 24, 25]. However, in N2a cells, a murine neuroblastoma cell line, the increased phosphorylation of CREB by GBE was not dependent on the PKA signalling pathway, a direct upstream regulator of CREB [12]. Furthermore, GBE reduced the hyperphosphorylation of tau at Ser262, Ser404, Ser396 and Thr231 sites as well as rescued the activity of tau kinase (GSK3β) in a rat model of AD [26]. Thus, the neuroprotective mechanism(s) of GBE could be mediated via multiple pathways such as by an indirect effect via inhibiting glycogen synthase kinase 3 (GSK3) involved in the PI3K/AKT/mTOR pathway that is related to neurogenesis, proliferation and migration [27, 28]. As we demonstrated, the Akt/mTOR signalling pathway also seems to play an important role in the effect of GBE on neurite outgrowth of human neuroblastoma cells (SH-SY5Y cells). Fig 6 summarizes the effects of GBE on this specific pathway. GBE induced phosphorylation of IGF-1R receptor (Tyr1135/Tyr1136) which invokes downstream changes in the metabolic pathway (Fig 6) [29]. This may trigger a signaling transduction cascade to the cell nucleus thus modulating some cellular functions such as promotion of cell survival via PI3K/Akt (Fig 6) [29]. These act in a coordinated manner to regulate lipid, glucose, and protein metabolism. PIP3 is produced by PI3K-dependent phosphorylation of PIP2, which in turn activates Akt and mammalian target of rapamycin (mTOR) [30]. The reverse and antagonizing reaction is regulated by the phosphatase PTEN (phosphatase and tensin homolog) [31, 32]. In the nervous system, PTEN inhibition leads to stem cell proliferation, neurite outgrowth and dendritic spine maturation [30, 33].

Notably, we could show that GBE was able to increase the inactivation of PTEN (phosphorylation at Ser380) which activates Akt (phosphorylation at Ser473) and mTOR (phosphorylation at Ser2448). In parallel, GBE inhibited signaling molecules like GSK3 [30]. Indeed, the activated form of Akt influences protein synthesis via mTOR as well as glucose metabolism via GSK3. We detected an increase of phosphorylated GSK3β (at Ser9), which is known to play a

central role in numerous cellular processes, including metabolism, cell-cycle and survival (Fig 6).

Akt activates the mTOR pathway via phosphorylation of tuberous sclerosis protein 2 (TSC2). GBE was able to increase the inactive form of TSC2 at the ser939 site as well as the active form of mTOR (Fig 6) [34]. These findings suggest that GBE might promote neurite growth via the PI3K/AKT/mTOR signaling pathway that potentially modulates neuronal plasticity. Therefore and in line with previous reports [12, 13], we propose that GBE is able to increase neurotrophic factors like BDNF, which also seems to be able to trigger the cascade activating the PI3K/AKT/mTOR signaling pathway and inhibiting PTEN and GSK3β (Fig 6). Further experiments are needed to dissect the exact underlying mechanisms correlating neurite outgrowth and the activation of AKT/mTOR signaling, for example by using an inhibitor of AKT (e.g. MK2206), and to detect the potential changes in the respective downstream signals [35]. Together with the protective effects of GBE on mitochondrial function [7], this might lead to neuronal survival, neuroplasticity and neurite outgrowth.

In conclusion, the present study demonstrates that GBE can markedly promote neurite outgrowth and activates the phosphorylation of Akt, GSK3β, IGF1R, mTOR, TSC2 and PTEN. In line with the already-mentioned mitochondrial function improving properties of GBE, it might represent one possible important pathway contributing to the initiation of neurite protrusion and subsequent elongation.

## Supporting information

**S1 Fig. Original pictures (confocal microscope, x10, immunostaining (IMS) with βIIItubuline/ Alexa488) of a 2D cell culture used for analysis using ImageJ software (Figs 1 and 2).** (TIF)

**S2 Fig. Original pictures (confocal microscope, x10, immunostaining (IMS) with βIIItubuline/ Alexa488) of a 3D-matrix cell culture used for analysis using ImageJ software (Figs 3 and 4).** (TIF)

**S1 Table. The data are presented as mean (M), standard deviation (±SD) corresponding to the number of cells (N).** Values are normalized on 100% of the control group (untreated CTRL, N = 108). (TIF)

**S2 Table. The data are presented as mean (M), standard deviation (±SD) corresponding to the number of cells (N).** Values are normalized on 100% of the control group (untreated CTRL, N = 112). (TIF)

**S3 Table. The Median Fluorescence Intensity (MFI) values are presented as mean (M), standard deviation (±SD) and number of replicates (n).** (TIF)

## Author Contributions

**Investigation:** Imane Lejri.

**Supervision:** Anne Eckert.

**Writing – original draft:** Imane Lejri, Amandine Grimm.

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
