## [Decision Letter · Decision Letter 0]

31 Jul 2019

PONE-D-19-18727

Ginkgo biloba extract increases neurite outgrowth via activation of the Akt/mTOR pathway

PLOS ONE

Dear Prof. Dr. ECKERT,

Thank you for submitting your manuscript to PLOS ONE. After careful consideration, we feel that it has merit but does not fully meet PLOS ONE’s publication criteria as it currently stands. Therefore, we invite you to submit a revised version of the manuscript that addresses the points raised during the review process.

We would appreciate receiving your revised manuscript by Sep 30, 2019. To enhance the reproducibility of your results, we recommend that if applicable you deposit your laboratory protocols in protocols.io, where a protocol can be assigned its own identifier (DOI) such that it can be cited independently in the future. For instructions see: http://journals.plos.org/plosone/s/submission-guidelines#loc-laboratory-protocols

We look forward to receiving your revised manuscript.

Kind regards,

Wenhui Hu, M.D., Ph.D.

Academic Editor

PLOS ONE

Journal Requirements:

https://www.thieme-connect.de/products/ejournals/abstract/10.1055/s-0029-1240204

https://www.sciencedirect.com/science/article/pii/S0149763414000256?via%3Dihub

In your revision ensure you cite all your sources (including your own works), and quote or rephrase any duplicated text outside the methods section. Further consideration is dependent on these concerns being addressed."

This study was supported by a principal investigator- initiated research grant supported by Vifor SA, Switzerland. 

We note that you received funding from a commercial source: Vifor SA, Switzerland.

Reviewers' comments:

Reviewer's Responses to Questions

**Comments to the Author**

1. Is the manuscript technically sound, and do the data support the conclusions?

Reviewer #1: Partly

Reviewer #2: Yes

2. Has the statistical analysis been performed appropriately and rigorously? 

Reviewer #1: Yes

Reviewer #2: Yes

3. Have the authors made all data underlying the findings in their manuscript fully available?

Reviewer #1: Yes

Reviewer #2: Yes

4. Is the manuscript presented in an intelligible fashion and written in standard English?

Reviewer #1: Yes

Reviewer #2: Yes

5. Review Comments to the Author

Reviewer #1: This manuscript by Anne Eckert and colleagues investigate the effect of the Standardized Ginkgo biloba extract (GBE) LI1370 on neuroplasticity and neurite outgrowth using SH-SY5Y neuroblastoma cells in a 2D and 3D surface culture. The authors evaluated parameters of neurite outgrowth and showed that GBE (10 and 100 µg/ml) significantly increased neurite outgrowth after 3 days of treatment with a comparable effect than that NGF. The investigators also showed an increase of phosphorylated IGF1R (Tyr1135/Tyr1136), Akt (Ser473), TSC2 (Ser939), mTOR (Ser2448), PTEN (Ser380) and GSK3β (Ser9) after the treatment of GBE.

Although the role of GBE in promoting dendritic growth has been confirmed by others, the mechanism is not clear and it’s attractive to explore the underlying pathways of GBE effect on the neurite outgrowth. The Luminex technology used in this manuscript is interesting, which could quantitatively detect each protein. I may have missed something important, and I was confused how the investigators drawn the conclusion that GBE promotes neurite growth via activation of the PI3K/Akt/mTOR pathway. It may be appropriate to Inhibit or enhance a site in the pathway and detect changes in downstream signals.

Minor comments:

1. Please add a citation for this sentence “By generating energy, regulating subcellular calcium and redox homeostasis, mitochondria play an important role in controlling fundamental processes in neuroplasticity, including neural differentiation, neurite outgrowth, neurotransmitter release and dendritic remodeling”.

2. Please confirm whether reference 9, 10 and 24 were incorrectly cited.

3. Please confirm reference 27 has been cited at corresponding location in this manuscript.

Reviewer #2: The study is intended to investigate the intracellular signal transduction pathways involved in promoting the neuroplasticity which is targeted by GBE. The topic is a hot spot at present.

See attached file for minor linguistic errors

6. PLOS authors have the option to publish the peer review history of their article (what does this mean?). If published, this will include your full peer review and any attached files.

Reviewer #1: No

Reviewer #2: No

---

## [Author Response · Author response to Decision Letter 0]

10 Oct 2019

Dear editors and dear reviewers,

Thank you very much for the time you took to read and review our manuscript as well as for your valuable comments which motivated us to improve our work. We are grateful for your advice and we revised our manuscript accordingly. We tried to address your comments in the best way. You can see all the changes throughout the manuscript marked in green. We changed one by one all of the requested modifications that were suggested by the reviewer 1 and we corrected the linguistic errors identified by the reviewer 2. We also adapted the title in accordance to the involvement of the Akt/ mTOR pathway discussed in the manuscript.

Especially, we modified the following:

JOURNAL REQUIREMENTS:

http://www.journals.plos.org/plosone/s/file?id=wjVg/PLOSOne_formatting_sample_main_body.pdf

http://www.journals.plos.org/plosone/s/file?id=ba62/PLOSOne_formatting_sample_title_authors_affiliations.pdf

Response: We re-organized the manuscript according to the journal PLOS ONE's style requirements including title, author, affiliations and the manuscript body formatting guidelines. We shorted the corresponding author section with only the email as required. We modified also the figure citations in the whole manuscript (pages 10 and 11, 16 and 17). 

https://www.thieme-connect.de/products/ejournals/abstract/10.1055/s-0029-1240204

https://www.sciencedirect.com/science/article/pii/S0149763414000256?via%3Dihub

In your revision ensure you cite all your sources (including your own works), and quote or rephrase any duplicated text outside the methods section. Further consideration is dependent on these concerns being addressed."

Response: We completely agree with the editors. We added some references from our own work (page 3 and 5). The section (pages 3) was shortened and rephrased.

This study was supported by a principal investigator- initiated research grant supported by Vifor SA, Switzerland. 

We note that you received funding from a commercial source: Vifor SA, Switzerland.

Response: The study was only partially supported by Vifor SA, Switzerland. To avoid any misunderstanding, the acknowledgement was removed from the manuscript.

Please now state:

Financial disclosure: This study was partly supported by a principal investigator (AE) - initiated research grant supported by Vifor Pharma Switzerland, with regard to consumables and materials as well as GBE supply. Remaining funding was provided by grants from the Swiss National Science Foundation (SNF #31003A_149728, to AE) and Synapsis Foundation - Alzheimer Research Switzerland ARS to AG, and funding from the Transfaculty Research Platform, Molecular & Cognitive Neuroscience, University of Basel. The funders had no role in study design, data collection and analysis, decision to publish, or preparation of the manuscript.

Competing Interests Statement: The authors have read the journal’s policy and have the following conflicts: AE received study grants from Vifor Pharma Switzerland. Moreover AE and AG received lecturer/consultant fees from Vifor Pharma Switzerland. This does not alter our adherence to PLOS ONE policies on sharing data and materials.

REVIEWER 1: 

This manuscript by Anne Eckert and colleagues investigate the effect of the Standardized Ginkgo biloba extract (GBE) LI1370 on neuroplasticity and neurite outgrowth using SH-SY5Y neuroblastoma cells in a 2D and 3D surface culture. The authors evaluated parameters of neurite outgrowth and showed that GBE (10 and 100 µg/ml) significantly increased neurite outgrowth after 3 days of treatment with a comparable effect than that NGF. The investigators also showed an increase of phosphorylated IGF1R (Tyr1135/Tyr1136), Akt (Ser473), TSC2 (Ser939), mTOR (Ser2448), PTEN (Ser380) and GSK3β (Ser9) after the treatment of GBE.

Although the role of GBE in promoting dendritic growth has been confirmed by others, the mechanism is not clear and it’s attractive to explore the underlying pathways of GBE effect on the neurite outgrowth. The Luminex technology used in this manuscript is interesting, which could quantitatively detect each protein. I may have missed something important, and I was confused how the investigators drawn the conclusion that GBE promotes neurite growth via activation of the PI3K/Akt/mTOR pathway. It may be appropriate to Inhibit or enhance a site in the pathway and detect changes in downstream signals.

Response: We thank the reviewer 1 for this valuable comment. We thoroughly state in the revised manuscript version that the effect of GBE on neurite outgrowth possibly involves the activation of the PI3K/Akt/mTOR pathway (Pages 13 to 15). We demonstrated in our manuscript that GBE is able to improve the neurite outgrowth and to increase the phosphorylated protein involved the PI3K/Akt/mTOR pathway. We state that the PI3K/Akt/mTOR signaling pathway seems to play an important role in the effects of GBE on neurite outgrowth (pages 13 and page 14), and that the activation of this pathway may represent one possible mechanism.

In addition, we completely agree with the reviewer 1 that further experiments are needed to dissect the exact underlying mechanisms correlating neurite outgrowth with the activation of PI3K/Akt/mTOR pathway, by using “as suggested by the reviewer”, specific inhibitors, e.g: AKT inhibitor, MK2206 (Yu et al 2018 PMID: 30098550) to see if a treatment with GBE still induces the expansion of the neurite as well as activates the respective downstream signal (page 14). 

Again, we thank the reviewer for this suggestion that we keep in mind for our further investigations. Accordingly, the section (pages 13 to 15) and the title were modified.

Modified title: Ginkgo biloba extract increases neurite outgrowth and activates the Akt/mTOR pathway

Minor comments:

1. Please add a citation for this sentence “By generating energy, regulating subcellular calcium and redox homeostasis, mitochondria play an important role in controlling fundamental processes in neuroplasticity, including neural differentiation, neurite outgrowth, neurotransmitter release and dendritic remodeling”.

Response: You are right, thank you. The sentence (page 3) was rephrased and the corresponding reference was added (Cheng et al 2010, PMID: 20957078)

2. Please confirm whether reference 9, 10 and 24 were incorrectly cited.

Response: We apologize for these errors and we confirm that the references 9, 10 and 24 were incorrectly cited (pages 12 and 14). We corrected and replaced with the right citation respectively (pages 12 and 14).

3. Please confirm reference 27 has been cited at corresponding location in this manuscript.

Response: Thank you for noticing our omission. We added the reference 27 (Cunha et al PMID: 20162032) at the corresponding location page 11 of our manuscript.

REVIEWER 2:

The study is intended to investigate the intracellular signal transduction pathways involved in promoting the neuroplasticity which is targeted by GBE. The topic is a hot spot at present.

See attached file for minor linguistic errors.

Response: We thank the reviewer 2 for this concern and for his/her corrections. We included his/her modifications about the general grammar and language improvement in the revised manuscript.

We thank the reviewers for their constructive criticism and most helpful comments that definitively helped to improve our manuscript. We hope that with the changes made, the manuscript is now suitable for publication.

---

## [Decision Letter · Decision Letter 1]

13 Nov 2019

Ginkgo biloba extract increases neurite outgrowth and activates the Akt/mTOR pathway

PONE-D-19-18727R1

Dear Dr. ECKERT,

We are pleased to inform you that your manuscript has been judged scientifically suitable for publication and will be formally accepted for publication once it complies with all outstanding technical requirements.

With kind regards,

Wenhui Hu, M.D., Ph.D.

Academic Editor

PLOS ONE

Additional Editor Comments (optional):

Reviewers' comments:

Reviewer's Responses to Questions

**Comments to the Author**

1. If the authors have adequately addressed your comments raised in a previous round of review and you feel that this manuscript is now acceptable for publication, you may indicate that here to bypass the “Comments to the Author” section, enter your conflict of interest statement in the “Confidential to Editor” section, and submit your "Accept" recommendation.

Reviewer #1: All comments have been addressed

Reviewer #2: All comments have been addressed

2. Is the manuscript technically sound, and do the data support the conclusions?

Reviewer #1: Yes

Reviewer #2: Yes

3. Has the statistical analysis been performed appropriately and rigorously? 

Reviewer #1: Yes

Reviewer #2: Yes

4. Have the authors made all data underlying the findings in their manuscript fully available?

Reviewer #1: Yes

Reviewer #2: Yes

5. Is the manuscript presented in an intelligible fashion and written in standard English?

Reviewer #1: Yes

Reviewer #2: Yes

6. Review Comments to the Author

Reviewer #1: The present study demonstrated that GBE could markedly promote neurite outgrowth and activited the phosphorylation of Akt, GSK3β, IGF1R, mTOR,TSC2 and PTEN，which might represent one possible important pathway. I am glad the authors have adopted my recommendations and modified the corresponding revisions to make the manuscript more rigorous.

Reviewer #2: The authors carefully addressed all raised points by reviewers

Therefore, No further comments

It's now worthy of publication

7. PLOS authors have the option to publish the peer review history of their article (what does this mean?). If published, this will include your full peer review and any attached files.

Reviewer #1: No

Reviewer #2: No

---

## [Editor Report · Acceptance letter]

18 Nov 2019

PONE-D-19-18727R1 

Ginkgo biloba extract increases neurite outgrowth and activates the Akt/mTOR pathway 

Dear Dr. ECKERT:

I am pleased to inform you that your manuscript has been deemed suitable for publication in PLOS ONE. Congratulations! Your manuscript is now with our production department. 

With kind regards,

on behalf of

Dr. Wenhui Hu 

Academic Editor

PLOS ONE